# The Effect of Adjuvant Chemotherapy on Localized Extraskeletal Osteosarcoma: A Systematic Review

**DOI:** 10.3390/cancers14102559

**Published:** 2022-05-23

**Authors:** Shinji Tsukamoto, Andreas F. Mavrogenis, Lucia Angelelli, Alberto Righi, Giuseppe Filardo, Akira Kido, Kanya Honoki, Yuu Tanaka, Yasuhito Tanaka, Costantino Errani

**Affiliations:** 1Department of Orthopaedic Surgery, Nara Medical University, 840, Shijo-Cho, Kashihara 634-8521, Japan; sh104@naramed-u.ac.jp (S.T.); kahonoki@naramed-u.ac.jp (K.H.); yatanaka@naramed-u.ac.jp (Y.T.); 2First Department of Orthopaedics, School of Medicine, National and Kapodistrian University of Athens, Athens 15562, Greece; 3Applied and Translational Research Center, IRCCS Istituto Ortopedico Rizzoli, 40136 Bologna, Italy; lucia.angelelli@ior.it (L.A.); giuseppe.filardo@ior.it (G.F.); 4Department of Pathology, IRCCS Istituto Ortopedico Rizzoli, Via di Barbiano 1/10, 40136 Bologna, Italy; alberto.righi@ior.it; 5Department of Rehabilitation Medicine, Nara Medical University, 840, Shijo-Cho, Kashihara 634-8521, Japan; akirakid@naramed-u.ac.jp; 6Department of Rehabilitation Medicine, Wakayama Professional University of Rehabilitation, 3-1, Minamoto-Cho, Wakayama 640-8222, Japan; tanakayuu717@gmail.com; 7Department of Orthopaedic Oncology, IRCCS Istituto Ortopedico Rizzoli, Via Pupilli 1, 40136 Bologna, Italy; costantino.errani@ior.it

**Keywords:** chemotherapy, extraskeletal osteosarcoma, prognosis, surgery, survival

## Abstract

**Simple Summary:**

The effect of (neo) adjuvant chemotherapy on localized extraskeletal osteosarcomas (ESOS) is controversial. We conducted a systematic review of studies comparing 5-year disease-free survival between patients who underwent surgery combined with (neo) adjuvant chemotherapy and those who underwent surgery alone for localized ESOS. The 5-year disease-free survival rate in the surgery and (neo)adjuvant chemotherapy group was 47.9% (187 of 390 patients) and the 5-year disease-free survival rate in the surgery-only group was 40.4% (150 of 371 patients). The overall pooled odds ratio was 1.23 (95% confidence interval, 0.69 to 2.19; *p* = 0.479). The effect of adjuvant chemotherapy on localized ESOS may be limited. Therefore, routine adjuvant chemotherapy for localized ESOS should be avoided.

**Abstract:**

(1) Background: Extraskeletal osteosarcoma (ESOS) is a malignant tumor characterized by the production of bone or bone matrix by tumor cells without any continuity into the skeletal bones. The standard treatment for localized ESOS is wide resection; however, the effect of (neo)adjuvant chemotherapy remains unclear. To investigate the effect of (neo)adjuvant chemotherapy for localized ESOS, we conducted a systematic review of studies comparing the 5-year disease-free survival rate between patients who underwent surgery combined with (neo)adjuvant chemotherapy and those who underwent surgery alone. (2) Methods: Of the 210 studies identified by systematically searching the PubMed, Embase, and Cochrane Central Register of Controlled Trials databases, 12 were included in the final analysis. These 12 articles were not randomized controlled trials, but retrospective studies. In total, 761 patients with localized ESOS were included in this study. (3) Results: The 5-year disease-free survival rate was 47.9% (187 of 390 patients) in the surgery and (neo)adjuvant chemotherapy group and 40.4% (150 of 371 patients) in the surgery alone group. The overall pooled odds ratio was 1.23 (95% confidence interval, 0.69–2.19; *p* = 0.479) and the heterogeneity I^2^ was 37%. (4) Conclusions: The effect of adjuvant chemotherapy on localized ESOS seems to be limited. Therefore, routine use of adjuvant chemotherapy for localized ESOS should be avoided. However, further randomized controlled trials are required to confirm these results.

## 1. Introduction

Extraskeletal osteosarcoma (ESOS) is a malignant tumor characterized by the production of bone or bone matrix by tumor cells that occur without bone continuity [1]. ESOS accounts for less than 1% of all soft tissue sarcomas and approximately 4% of all osteosarcomas. ESOS often occurs in patients in their 50′s to 70′s and men are more frequently affected than women [1]. Histologically, tumors are composed of spindle or polygonal cells with variously pleomorphic and cytologically atypical mitotic figures. Tumor necrosis is a common feature. Identification of neoplastic bone intimately associated with tumor cells, which may be deposited in lace-like, trabecular, or sheet-like patterns, is necessary to obtain an accurate diagnosis (Figure 1) [1].

Patients with localized ESOS account for 81–84% of all patients with ESOS [2,3]. The standard treatment for localized ESOS is wide resection, but the efficacy of adjuvant chemotherapy remains controversial [4]. According to the ESMO-EURACAN-GENTURIS-ERN PaedCan Clinical Practice Guideline for the diagnosis, treatment, and follow-up of patients with soft tissue sarcomas, ESOS is considered a high-grade soft tissue sarcoma with questionable clinical similarity to osteosarcoma and adjuvant chemotherapy provides limited benefit [5]. In contrast, according to the National Comprehensive Cancer Network guidelines, ESOS can be treated with adjuvant chemotherapy using ifosfamide or platinum-based therapy (cisplatin/doxorubicin) [6]. In addition, the appropriate chemotherapy regimen for bone or soft tissue sarcomas remains controversial [2,7,8,9]. Because ESOS rarely occurs, only retrospective studies have been conducted and no randomized controlled trials (RCTs) have examined the effect of (neo)adjuvant chemotherapy on localized ESOS. Therefore, the effect of (neo)adjuvant chemotherapy on localized ESOS remains unclear. To investigate the effect of (neo)adjuvant chemotherapy on localized ESOS, we performed a systematic review of studies comparing the 5-year disease-free survival between patients who underwent surgery alone and those who underwent surgery and (neo)adjuvant chemotherapy for localized ESOS.

## 2. Methods

The study results were reported in accordance with the Preferred Reporting Items for Systematic Reviews and Meta-analyses 2020 statement [10]. This study was registered in the UMIN Clinical Trials Registry as UMIN000047442 (http://www.umin.ac.jp/ctr/index.htm (accessed on 8 April 2022)).

### 2.1. Eligibility Criteria

Only studies reporting the 5-year disease-free survival rates of patients who underwent surgery alone and those who underwent surgery and (neo)adjuvant chemotherapy for localized ESOS were included. (i) Patients with distant metastases at presentation were excluded from the study. Studies that did not specify the 5-year disease-free survival rate, did not have a control group, or had fewer than five patients were also excluded. (ii) Patients who underwent surgery alone for localized ESOS and received chemotherapy for distant metastases that developed during the disease course were classified into the surgery alone group. (iii) Only studies published in English, Italian, or Japanese were included and the year of publication was not restricted. Only human studies were included (animal studies were excluded).

### 2.2. Literature Search and Study Selection

Relevant literature was systematically searched in the PubMed, Embase, and Cochrane Central Register of Controlled Trials (CENTRAL) databases on 26 March 2022 (Appendix A). In addition, the bibliographies of the retrieved literature were used to identify other relevant studies. Publication bias was assessed using funnel plots and Egger’s test.

### 2.3. Data Collection and Presentation

Two authors (ST and LA) independently selected the studies and extracted the data. In cases of disagreement, an agreement was reached between the two authors, or a third author was consulted. The following data were collected using a data-collection sheet: (i) baseline data (author, year of publication, type of study, follow-up period since diagnosis of ESOS, and number of patients with localized ESOS); (ii) number of patients who underwent surgery and (neo)adjuvant chemotherapy for localized ESOS, number of patients who remained disease-free within a 5-year period, number of patients who underwent surgery alone for localized ESOS, and number of patients who remained disease-free within a 5-year period; (iii) ratio of male/female patients, age, tumor site, tumor size, surgical margin, adjuvant radiotherapy, and histological grade in the surgery plus (neo)adjuvant chemotherapy and surgery alone groups; and (iv) chemotherapy regimen and histological evidence of necrosis following preoperative chemotherapy.

### 2.4. Data Summary, Synthesis, and Meta-Analysis

The data extracted from the collected study data were summarized (Table 1 and Table 2). The dataset included the name of the first author, year of publication, number of patients who underwent surgery, (neo)adjuvant chemotherapy for localized ESOS, number of patients who remained disease-free within a 5-year period, number of patients who underwent surgery only for localized ESOS, and the number of patients who remained disease-free within a 5-year period. Random effects models were used to estimate the odds ratios (ORs) to compare the 5-year disease-free survival rates between the surgery plus (neo) adjuvant chemotherapy and surgery alone groups. The extent of heterogeneity between the studies was evaluated using the inconsistency statistic (I^2^). All statistical analyses were performed assuming a two-sided test at a 5% significance level using ProMeta software version 3 (Internovi di Scarpellini Daniele s.a.s.).

### 2.5. Assessment of Methodological Quality

Two authors (ST and LA) independently assessed the quality of all included studies. Disagreements were resolved through discussion between the two authors or consultation with a third author. The studies included in the final analysis were independently assessed according to the Risk of Bias in Non-Randomized Studies (RoBANS) tool to assess the quality of non-randomized studies included in meta-studies [20].

### 2.6. Search Results

Of the 580 studies identified during the database search, 12 were finally included in this study (Figure 2, Table 1 and Table 2) [2,3,8,11,12,13,14,15,16,17,18,19]. None of the 12 studies were RCTs. Funnel plots of the 5-year disease-free survival rates were symmetrical (Figure 3). Funnel plots were constructed with a 95% confidence interval. However, in a study by Longhi et al., the number of patients was the second highest and the effect size was the third highest; therefore, it was outside the 95% confidence interval (Figure 3) [2]. The results of Egger’s test had a *p* value of 0.577. Therefore, publication bias was not observed.

### 2.7. Demographic Data and Ratio of the Patients Who Underwent Surgery and Adjuvant Chemotherapy or Surgery Alone

A total of 761 patients with localized ESOS were eligible for inclusion, of which 390 (51.2%) underwent surgery and (neo)adjuvant chemotherapy, while 371 (48.8%) underwent surgery alone (Table 1).

### 2.8. Methodological Quality of the Included Studies

The quality of the individual studies was assessed using the RoBANS tool and an overall moderate risk of bias was observed. All 12 included studies showed that the quality of the “selection of participants” was high, “confounding variables” was high, “measurement of exposure” was low, “blinding of outcome” was low, “incomplete outcome data” was low, and “selective outcome reporting” was low.

## 3. Results

In patients with localized ESOS, the 5-year disease-free survival rate was similar between the surgery plus (neo)adjuvant chemotherapy and surgery alone groups. The 5-year disease-free survival rates in the surgery plus (neo)adjuvant chemotherapy and surgery alone groups were 47.9% (187 of 390 patients) and 40.4% (150 of 371 patients), respectively. The overall pooled OR was 1.23 (95% confidence interval, 0.69–2.19; *p* = 0.479) and the heterogeneity I^2^ was 37% (Figure 4) (Table 1).

The proportion of male participants was 0–100% in the surgery and (neo)adjuvant chemotherapy group and 50–75% in the surgery alone group [3,8,13,14,16,17,19]. The patients’ mean ages were 15–58 years in the surgery and (neo)adjuvant chemotherapy group and 48–67 years in the surgery alone group, with the surgery alone group being older [3,8,13,14,16,17,19]. Deep-seated tumors were detected in 87–100% of patients in the surgery and (neo)adjuvant chemotherapy group, while they were detected in 63–84% of patients in the surgery alone group [3,16,17]. Tumors located in the trunk were only detected in 0–46% of patients in the surgery and (neo)adjuvant chemotherapy group, while they were detected in 0–100% of patients in the surgery alone group [8,12,13,14,16,17,19]. The mean tumor size in the surgery and (neo)adjuvant chemotherapy group ranged from 2 to 12 cm, while it was 8 to 12 cm in the surgery alone group [3,13,14,16,17]. Approximately 85–100% of the patients in the surgery and (neo)adjuvant chemotherapy group had a R0 surgical margin, while 0–100% of those in the surgery alone group had an R0 surgical margin [3,8,12,19]. The proportion of patients in the surgery and (neo) adjuvant chemotherapy group who received adjuvant radiation therapy ranged from 50 to 85%, while the proportion in the surgery alone group ranged from 31 to 75% [13,19]. Moreover, 75–100% of the total patients had histologically high-grade tumors [2,3,8,14,15,16,17,18]. With regard to the chemotherapy regimens, the osteosarcoma chemotherapy regimen was used in 48–58% of patients, while the soft tissue sarcoma chemotherapy regimen was used in 33–36% of patients [2,3]. Approximately 19–50% of patients had >90% tumor necrosis after neoadjuvant chemotherapy (Table 2) [11,18].

## 4. Discussion

The efficacy of (neo)adjuvant chemotherapy for localized ESOS remains controversial. We collected and analyzed studies that compared the 5-year disease-free survival rate between patients with localized ESOS treated with surgery and (neo)adjuvant chemotherapy and those treated with surgery alone. To date, no systematic review of the literature has investigated the effect of (neo)adjuvant chemotherapy on localized ESOS. The Surveillance Epidemiology and End Results database contains as many as 310 patients with ESOS, but lacks information on the use of adjuvant chemotherapy [21]. No difference was observed in the 5-year disease-free survival rate between the surgery plus (neo)adjuvant chemotherapy and surgery alone groups. Therefore, the effect of (neo)neoadjuvant chemotherapy on localized ESOS seems limited.

This study had some limitations. All studies included in this systematic review were retrospective in nature and their results were potentially biased by the confounding factors of adjuvant chemotherapy. Adjuvant chemotherapy is often administered to young patients [3,8,13,14,16,17,19] and those with deep-seated tumors [3,16,17]. Older patients with ESOS have a poorer prognosis [2,3,18,22]. The prognosis for deep-seated ESOS is poorer than the prognosis for superficially located ESOS [3,7,9,15,23,24]. RCTs can avoid many of these biases by randomly allocating participants to the study groups. However, well-designed cohort and observational studies with strong effects may provide more reliable information. Second, various chemotherapy regimens are available, however, most are osteosarcoma-type or soft tissue sarcoma-type regimens. Osteosarcoma-type chemotherapy corresponds to a multi-drug regimen similar to that used for bone osteosarcoma, with cisplatin, doxorubicin, ifosfamide, and methotrexate. Soft tissue sarcoma-type chemotherapy is a regimen similar to that used for soft tissue sarcomas (anthracycline with or without ifosfamide) [2]. Therefore, all regimens use conventional cytotoxic anticancer drugs and do not contain new molecular-targeted drugs or immune checkpoint inhibitors. Third, two large cohort studies by Longhi et al. and Heng et al. showed opposite results [2,3]. Adjuvant chemotherapy was more frequently performed for patients with deep, large tumors and a poor prognosis in a study by Heng et al. [3]. This may have made adjuvant chemotherapy less effective in the study by Heng et al.

The results of this study indicated that the efficacy of (neo)neoadjuvant chemotherapy for localized ESOS is limited. Ahmad et al. reported that among 27 patients with ESOS who received doxorubicin-based chemotherapy, the response rate was 19%, with two patients having a complete response and three patients having a partial response [9]. In a study conducted by Torigoe et al., of the eleven patients who received doxorubicin- and/or cisplatin-based chemotherapy, five achieved a partial response (45% response rate) [19]. Paludo et al. evaluated 11 patients with ESOS treated with neoadjuvant therapy and reported a response rate for platinum-containing regimens of 27% (3 of 11 patients) [7].

With regard to adjuvant chemotherapy regimens, it remains unclear whether ESOS should be treated using the regimen for high-grade soft tissue sarcoma or conventional osteosarcoma of the bone, as ESOS is derived from soft tissues but has similar histopathologic features to osteosarcoma [13,23]. Ahmad et al. favored doxorubicin-based chemotherapy over platinum (response rate: 19% (5 of 27 patients) vs. 13% (2 of 15 patients)) [9]. Wakamatsu et al. reported that patients treated with adjuvant chemotherapy consisting of doxorubicin and ifosfamide had significantly better 5-year disease-specific survival rates than those treated with other regimens (100% vs. 40%, *p* = 0.0327) [8]. In contrast, other studies have reported that platinum-containing chemotherapy was more effective [2,7]. Paludo et al. evaluated a cohort of 43 patients: 27 received chemotherapy, while the remaining patients (*n* = 22, 84%) received platinum-containing chemotherapy [7]. Patients treated with platinum-containing regimens showed better overall survival (*p* = 0.01) and progression-free survival (*p* = 0.007) than those treated with non-platinum regimens based on the results of the multivariate analysis. Recurrence rates were lower in the platinum-based group (41%, 9 of 22 patients) than in the non-platinum-based group (100%, 4 of 4 patients; *p* = 0.02) [7]. Longhi et al. reported that in an analysis of 211 patients with localized ESOS, 121 were treated with adjuvant chemotherapy, 69 (58%) were treated with an osteosarcoma regimen, and 43 (36%) were treated with a soft tissue sarcoma regimen. The osteosarcoma regimen was more effective than the soft tissue sarcoma regimen (5-year disease-free survival rate: 62% vs. 48%, respectively; *p* = 0.05) [2].

ESOS generally occurs at the age of 60 and has a superior prognosis compared to skeletal osteosarcomas, which occur in the same age group [22]. The prognosis of ESOS is similar to that of soft tissue sarcomas [25,26]. Our current treatment strategy for localized ESOS is similar to that used for other soft tissue sarcomas. Patients with localized disease are treated with wide resection alone.

## 5. Conclusions

The effect of adjuvant chemotherapy on localized ESOS appears to be limited. Therefore, routine use of adjuvant chemotherapy for localized ESOS should be avoided. However, further RCTs are required to confirm these results.

## Figures and Tables

**Figure 1 cancers-14-02559-f001:**
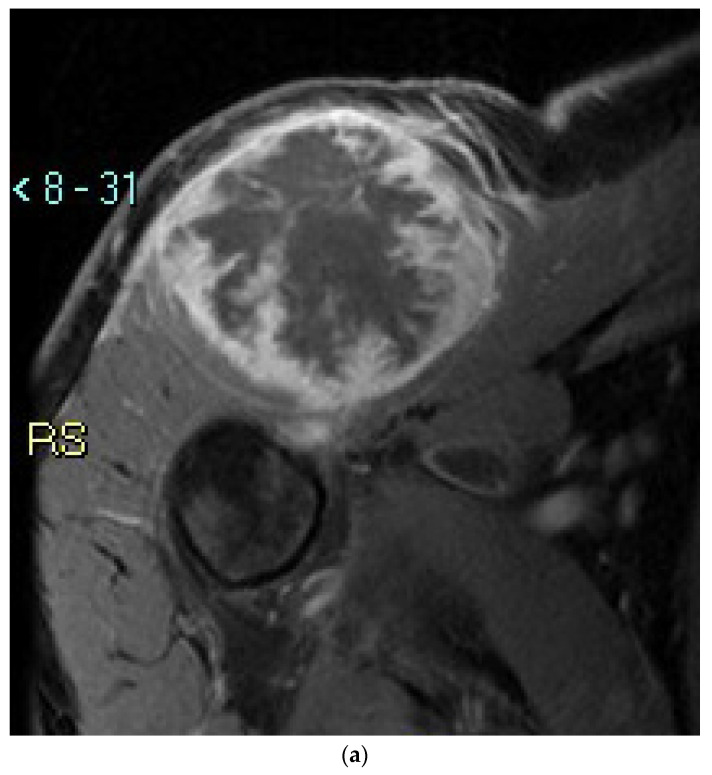
(**a**) Axial T1 fat-saturated MRI shows soft tissue mass with peripheral high signal and internal low signal in the muscle anterior to the right shoulder. (**b**) Axial contrast-enhanced CT scan shows a peripherally enhancing soft tissue mass located in the muscle, anterior to the right shoulder. Calcification is noted in the mass (arrow). (**c**) Macroscopically, the tumor is composed of tan−white tissue with gritty zones corresponding to bone formation. On hematoxylin and eosin, a neoplastic proliferation of atypical polygonal to spindle cells producing the malignant osteoid is evident ((**d**) 100× magnification, (**e**) 200× magnification). A strong immunohistochemical nuclear expression of SATB2 in neoplastic cells demonstrates the osteogenic differentiation and osteoid matrix production of the neoplasm ((**f**) 200× magnification).

**Figure 2 cancers-14-02559-f002:**
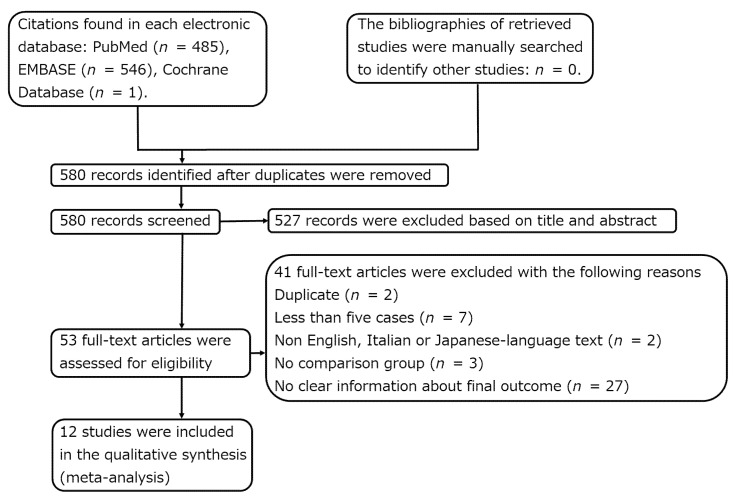
This flow chart shows the flow of database search used to identify relevant articles.

**Figure 3 cancers-14-02559-f003:**
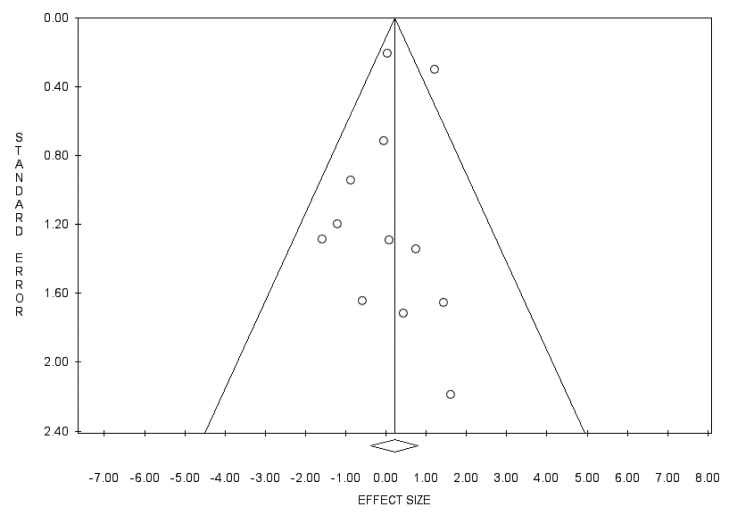
This funnel plot shows the process of detecting publication bias.

**Figure 4 cancers-14-02559-f004:**
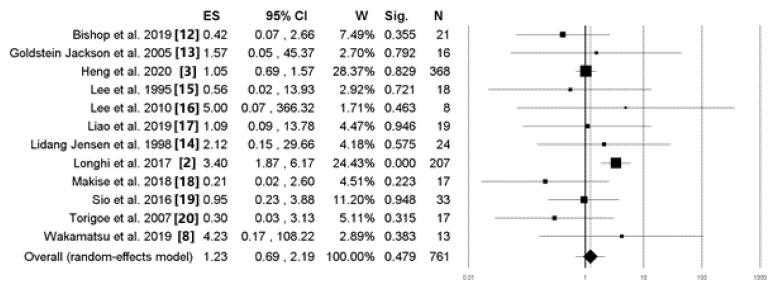
A forest plot shows the proportion of patients in the surgery combined with adjuvant chemotherapy and surgery groups who were disease-free for 5 years in the various studies. (ES: effect size (odds ratio); CI: confidence interval; W: weight; Sig: significance (*p*-value); N: total sample size).

**Table 1 cancers-14-02559-t001:** Overall study characteristics.

Study	Year	Type of Study	Follow-Up (Years)	Total Number of Patients with Localized ESOS	Number of Patients in the Surgery and Adjuvant Chemotherapy Group	Number of Patients who Were Disease-Free for 5 Years in the Surgery and Adjuvant Chemotherapy Group	Number of Patients in the Surgery Group	Number of Patients who Were Disease-Free for 5 Years in the Surgery Group
Bishop et al. [11]	2019	SR	Median, 10	21	7	3	14	9
Goldstein-Jackson et al. [12]	2005	MP	Median, 3.2	16	15	5	1	0
Heng et al. [3]	2020	MR	Median, 3	368	178	91	190	95
Lee et al. [13]	1995	SR	Mean, 5.9	18	2	0	16	4
Lee et al. [14]	2010	SR	NR	8	1	0	7	0
Liao et al. [15]	2019	SR	Median, 3	19	15	4	4	1
Lidang Jensen et al. [16]	1998	SR	Median, 1.3	24	5	1	19	2
Longhi et al. [2]	2017	MR	Median, 1.9	207	121	67	86	23
Makise et al. [17]	2018	SR	Median, 2.6	17	9	1	8	3
Sio et al. [18]	2016	MR	Median, 3.8	33	13	7	20	11
Torigoe et al. [19]	2007	MR	Mean, 3.8	17	13	3	4	2
Wakamatsu et al. [8]	2019	MR	Median, 50.5	13	11	5	2	0

SR, single institutional non-randomized retrospective study; MR, multi-institutional non-randomized retrospective study; MP, multi-institutional non-randomized prospective study; NR, not reported.

**Table 2 cancers-14-02559-t002:** Details of the patients included in this study.

Study	Percentage of Male Patients (Surgery + Adj Chemo vs. Surgery)	Age (Surgery + Adj Chemo vs. Surgery)	Percentage of Deeply Located Tumors (Surgery + Adj Chemo vs. Surgery)	Percentage of Tumors Located in the Trunk (Surgery + Adj Chemo vs. Surgery)	Tumor Size (cm) (Surgery + Adj Chemo vs. Surgery)	Proportion of Patient with an R0 Surgical Margin (Surgery + Adj Chemo vs. Surgery)	Percentage of Patients who Received Adjuvant Radiotherapy (Surgery + Adj Chemo vs. Surgery)	Histological Grade (Surgery + Adj Chemo vs. Surgery)	Chemotherapy Regimen	Assessment of Histological Response to Preoperative Chemotherapy
Bishop et al. [11]	NR	NR	NR	NR	Larger tumor of >5 cm was associated with chemotherapy use.	NR	NR	NR	NR	99% or more necrosis: 50%
Goldstein-Jackson et al. [12]	NR	NR	NR	20% vs. 100%	NR	93% vs. 0%	NR	NR	DOX, CDDP, IFO, MTX, VP16, CBDCA	NR
Heng et al. [3]	61% vs. 57%	Median, 55 vs. 64	87% vs. 75%	NR	Median, 8.8 vs. 8.1	86% vs. 83%	NR	High-grade: 100%	Osteosarcoma type: 48%, Soft tissue sarcoma type: 33%, Unknown: 19%	NR
Lee et al. [13]	100% vs. 56%	Mean, 36 vs. 56	NR	0% vs. 25%	Mean, 12 vs. 8.8	NR	50% vs. 31%	NR	NR	NR
Lee et al. [14]	0% vs. 57%	Mean, 15 vs. 67	NR	0% vs. 57%	Mean, 2 vs. 8	NR	NR	High-grade: 60%	NR	NR
Liao et al. [15]	NR	NR	NR	NR	NR	NR	NR	High-grade: 77%, Low-grade: 23%	DOX, CDDP, IFO, MTX	NR
Lidang Jensen et al. [16]	40% vs. 53%	Mean, 47 vs. 64	100% vs. 84%	20% vs. 26%	Mean, 7 vs. 11	NR	NR	High-grade: 100%	NR	NR
Longhi et al. [2]	NR	Adjuvant chemotherapy was administrated more frequently in patients younger than 65 years. >65 yrs: 21% vs. <65 yrs: 79%	NR	NR	NR	NR	NR	Most are high-grade	Osteosarcoma type: 58%, Soft tissue sarcoma type: 36%	NR
Makise et al. [17]	56% vs. 50%	Mean, 53 vs. 56	89% vs. 63%	44% vs. 63%	Mean, 8.7 vs. 12	NR	NR	High-grade: 100%	NR	NR
Sio et al. [18]	NR	NR	NR	NR	NR	NR	NR	High-grade: 84%, Intermediate-grade: 14%	Mitomycin, DOX, CDDP, IFO, MTX, VP16	90% or more necrosis: 19%
Torigoe et al. [19]	77% vs. 75%	Mean, 50 vs. 48	NR	46% vs. 25%	NR	85% vs. 50%	85% vs. 75%	NR	DTIC, DOX, CDDP, IFO, MTX, VP16, CBDCA, Taxol, THP, VDS	NR
Wakamatsu et al. [8]	64% vs. 50%	Mean, 58 vs. 63	NR	45% vs. 0%	Mean, 9.7 vs. NR	100% vs. 100%	NR	High-grade: 75%, Intermediate-grade: 25%	DOX/CDDP/ IFO/MTX/ VP16/EPI/ CBDCA	NR

CBDCA, carboplatin; CDDP, cisplatin; DOX, doxorubicin; DTIC, dacarbazine; EPI, epirubicin; EPI, epirubicin; IFO, ifosfamide; MTX, methotrexate; NR, not reported; THP, pirarubicin; VDS, vindesine; VP-16, etoposide.

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
