# Peer review of "The Effect of Adjuvant Chemotherapy on Localized Extraskeletal Osteosarcoma: A Systematic Review"

_cancers, 2022, doi:10.3390/cancers14102559_

Round 1

Reviewer 1 Report

The role of adjuvant chemotherapy for extraskeletal osteosarcoma remains unknown. Therefore, this review article is valuable to improve our clinical practices, and I think it is worth publishing. However, some points need to address before acceptance.

As the authors mentioned in a discussion, the effect of chemo regimens remains unclear. Are there any survival differences among patients who received adjuvant chemotherapy depending on chemo regimen types? The analysis might give some answers to the aforementioned clinical question.

Furthermore, the variety of chemo regimens among the 12 articles is also an intense limitation of this review. The authors should mention this point in the discussion.

This review includes two large cohorts with opposite conclusions regarding the role of adjuvant chemotherapy. Please address your thoughts regarding the reasons in the discussion.

Please provide explanations of abbreviations for the top row of figure 4.

Author Response

Responses to Reviewers’ comments

Dear Reviewers, thank you for your detailed and thoughtful comments regarding our manuscript. We believe that our paper has markedly improved as a result of your valuable comments and feedback.

Reviewer 1

Comment

The role of adjuvant chemotherapy for extraskeletal osteosarcoma remains unknown. Therefore, this review article is valuable to improve our clinical practices, and I think it is worth publishing. However, some points need to address before acceptance.

As the authors mentioned in a discussion, the effect of chemo regimens remains unclear. Are there any survival differences among patients who received adjuvant chemotherapy depending on chemo regimen types? The analysis might give some answers to the aforementioned clinical question.

Response

Regarding the chemotherapy regimen, there were only four studies comparing the prognosis between the osteosarcoma-type regimen and the soft tissue sarcoma-type regimen, as shown in lines 237-260 of “Discussion” section (Ahmad et al.[9], Wakamatsu et al.[8], Paludo et al.[7], Longhi et al.[2].). The outcomes were response rate, 5-year disease-specific survival, overall survival and progression-free survival, and 5-year disease-free survival rate, respectively, and it was not possible to perform statistical analysis because they were different.

Comment

Furthermore, the variety of chemo regimens among the 12 articles is also an intense limitation of this review. The authors should mention this point in the discussion.

Response

We added the following sentence as a limitation in the “Discussion” section; “Second, various chemotherapy regimens are available, however, most are osteosarcoma-type or soft-tissue sarcoma-type regimens. Osteosarcoma-type chemotherapy corresponds to a multidrug regimen similar to that used for bone osteosarcoma, with cisplatin, doxo-rubicin, ifosfamide, and methotrexate. Soft tissue sarcoma-type chemotherapy is a regi-men similar to that used for soft tissue sarcomas (anthracycline with or without ifosfamide) [2]. Therefore, all regimens are conventional cytotoxic anticancer drugs and do not contain new molecular-targeted drugs or immune checkpoint inhibitors.”

Comment

This review includes two large cohorts with opposite conclusions regarding the role of adjuvant chemotherapy. Please address your thoughts regarding the reasons in the discussion.

Response

We added the following sentence as a limitation in the “Discussion” section; “Third, two large cohort studies by Longhi et al. and Heng et al. showed opposite results [2,3]. Adjuvant chemotherapy was more frequently performed in deep, large tumors with poor prognosis in a study by Heng et al. [3]. This may have made adjuvant chemotherapy less effective in the study by Heng et al.”

Comment

Please provide explanations of abbreviations for the top row of figure 4.

Response

We added the following description to Figure legend; “(ES: effect size (odds ratio); CI: confidence interval; W: weight; Sig: significance (p-value); N: to-tal sample size).”

Reviewer 2 Report

This is an interesting and well conducted systematic review on the comparison of 2 different therapeutic strategies for localized extraskeletal osteosarcoma (ESOS).

  • The title indicates a systematic review on the effect of adjuvant chemotherapy on ESOS, but the topic here is to compare the 5-year DFS between 2 different therapeutic strategies (surgery alone vs surgery + chemotherapy), in localized ESOS only and not in all ESOS patients. The study of Qi L. et al. (The role of chemotherapy in extraskeletal osteosarcoma, Med Sci Monit, 2020; 26: e925107) can have a more general conclusion about the role of chemotherapy in ESOS as the inclusion criteria for patients are broader. Surprisingly, this study has not been mentioned in the discussion and should appear, as the conclusions are in agreement. Therefore, in the reviewed study the title should indicate the precise extent of the study.
  • Because the study is only limited to localized ESOS, a phrase in the introduction should mention the percentage of patients with ESOS that have only localized disease.
  • In the funnel plot, we observe that one of the studies with the highest number of patients is out of the diagram (probably the study from Longhi et al.). This should be briefly discussed.

Author Response

Reviewer 2

Comment

This is an interesting and well conducted systematic review on the comparison of 2 different therapeutic strategies for localized extraskeletal osteosarcoma (ESOS).

The title indicates a systematic review on the effect of adjuvant chemotherapy on ESOS, but the topic here is to compare the 5-year DFS between 2 different therapeutic strategies (surgery alone vs surgery + chemotherapy), in localized ESOS only and not in all ESOS patients. The study of Qi L. et al. (The role of chemotherapy in extraskeletal osteosarcoma, Med Sci Monit, 2020; 26: e925107) can have a more general conclusion about the role of chemotherapy in ESOS as the inclusion criteria for patients are broader. Surprisingly, this study has not been mentioned in the discussion and should appear, as the conclusions are in agreement. Therefore, in the reviewed study the title should indicate the precise extent of the study.

Response

We revised the title as you pointed out. We added the following sentence in the “Discussion” section; “The Surveillance Epidemiology and End Results database contains as many as 310 pa-tients with ESOS, but lacks information on the use of adjuvant chemotherapy [21].”

Comment

Because the study is only limited to localized ESOS, a phrase in the introduction should mention the percentage of patients with ESOS that have only localized disease.

In the funnel plot, we observe that one of the studies with the highest number of patients is out of the diagram (probably the study from Longhi et al.). This should be briefly discussed.

Response

We added the following sentence in the “Introduction” section; “Patients with localized ESOS account for 81–84% of all patients with ESOS [2,3].”

We wrote funnel plot with 95ï¼… confidence interval. The study by Longhi et al. had the second largest sample size and the third largest effect size, so it was out of 95% confidence interval. We added the following sentence in the “Methods” section; “Funnel plots were constructed with 95% confidence interval. However, in a study by Longhi et al., the number of patients was the second highest, and the effect size was the third highest; therefore, it was outside the 95% confidence interval (Figure 3)[2].”

Reviewer 3 Report

The authors have compiled a relevant review entitled "Effect of Adjuvant Chemotherapy on Extraskeletal Osteosarcoma"The methodology of acquiring the needed information has been deatailed.The data supports the claim that adjuvant chemotherapy on localized ESOS is limited and that surgery alone provides similar if not better results.

The review highlights the shortcomings of chemotherapy in ESOS and offers researchers and clinicians to investigate further in the understanding of the mechanisms of ESOS treatment.

Author Response

Reviewer 3

Comment

The authors have compiled a relevant review entitled "Effect of Adjuvant Chemotherapy on Extraskeletal Osteosarcoma"The methodology of acquiring the needed information has been detailed. The data supports the claim that adjuvant chemotherapy on localized ESOS is limited and that surgery alone provides similar if not better results. The review highlights the shortcomings of chemotherapy in ESOS and offers researchers and clinicians to investigate further in the understanding of the mechanisms of ESOS treatment.

Responses

Thank you for your detailed and thoughtful comments regarding our manuscript. We believe that our paper has markedly improved as a result of your valuable comments and feedback.
